# Integrating Sustainable Development and Children's Rights: A Case Study on Wales

**Rhian Croke** [1,*], **Helen Dale** [2], **Ally Dunhill** [3], **Arwyn Roberts** [2], **Malvika Unnithan** [4] and **Jane Williams** [5]

1   Hillary Rodham Clinton School of Law, Swansea University, Swansea SA2 8PP, UK
2   Lleisiau Bach/Little Voices, National Lottery People and Places Fund 2012-2020,
    Swansea and Bangor University, Swansea SA2 8PP, UK; H.M.Dale@hotmail.com (H.D.);
    a.b.roberts@bangor.ac.uk (A.R.)
3   Independent Consultant and Researcher, Kingston Upon Hull HU6 8TA, UK; allydunhill@gmail.com
4   Northumbria University Law School, Newcastle upon Tyne NE1 8ST, UK;
    malvika.unnithan@northumbria.ac.uk
5   Observatory on the Human Rights of Children, Swansea University, Swansea SA2 8PP, UK;
    jane.m.williams@swansea.ac.uk
*   Correspondence: rhiancrokehrw@gmail.com or 917266@swansea.ac.uk

**Abstract:** The global disconnect between the Sustainable Development Goals (SDGs) and the Convention on the Rights of the Child (CRC), has been described as 'a missed opportunity'. Since devolution, the Welsh Government has actively pursued a 'sustainable development' and a 'children's rights' agenda. However, until recently, these separate agendas also did not contribute to each other, although they culminated in two radical and innovative pieces of legislation; the Rights of Children and Young Persons (Wales) Measure (2013) and the Well-being and Future Generations (Wales) Act (2015). This article offers a case study that draws upon the SDGs and the CRC and considers how recent guidance to Welsh public bodies for implementation attempts to contribute to a more integrated approach. It suggests that successful integration requires recognition of the importance of including children in deliberative processes, using both formal mechanisms, such as local authority youth forums, pupil councils and a national youth parliament, and informal mechanisms, such as child-led research, that enable children to initiate and influence sustainable change.

**Keywords:** children's rights; sustainable development; children's participation in decision making; Wales; devolution

## 1. Introduction

The international agenda for children's rights and the agenda on sustainable development have rapidly evolved over the last 30 years, although in parallel and not symbiotically. Intuitively, they should complement each other but in practice there has been a dissonance or separation in their development (Kilkelly 2020). Children's rights are anchored in international law by the most globally ratified human rights treaty, the 1989 Convention on the Rights of the Child (Committee on the Rights of the Child 2020b). In the same decade, the provenance of sustainable development can be found in the 1987 Brundtland Report which defined sustainable development as '[d]evelopment that meets the needs of the present without compromising the ability of future generations to meet their own needs' (World Commission on Environment and Development 1987, Ch.2 para 1).

Since the end of the 1980s, discourse on development led to the Millennium Development Goals and, in 2015, the Sustainable Development Goals (SDGs) (UNDP n.d.a). The UN Committee on the Rights of the Child has developed an extensive body of jurisprudence through the publication of interpretative documents including some 25 General Comments (Committee on the Rights of the Child 2020a). The origins of the two agendas, children's rights in internationally enshrined legal rights and sustainable development in international development discourse, has contributed to this divergence (Kilkelly 2020).

Additionally, the chasm between the UN's development efforts and its work on human rights has served to accentuate the disconnect between the two agendas (Kilkelly 2020).

Kilkelly has described the failure to articulate the SDGs in terms of legal rights as a 'missed opportunity' (Kilkelly 2020, p. 378). Nolan warns of the 'clear and present danger' of the SDGs to human rights (Nolan 2019), noting that the implementation mechanisms for the SDGs are not child-rights sensitive or compliant (Nolan 2020). Similarly, Vandenhole has described how a children's rights approach to sustainable development is still very much under construction (Vandenhole 2019).

This article presents a potential pathway towards better alignment of policy and practice in sustainable development and children's rights. Using Wales as a case study (Denzin and Lincoln 2011), it focuses on the first two decades of the devolved government (1999–2020), and offers insights from analysis of law reform, policy and practice on children's rights and sustainable development. Case studies can be particularly useful for informing policy because processes, problems and programmes can be examined to bring about understanding that can affect and even improve practice (Yin 1989). Applying the in-depth learning from a case study can be applied to broader issues and the macro-sociological context (Johnson 2010). This case study seeks to offer reflections that provide transferable learning to other contexts. In Wales, one of the three devolved regions of the United Kingdom, sustainable development and children's rights have generated specific and radical legislative measures: the Rights of Children and Young Persons (Wales) and the The Well-Being and Future Generations Act (Wales). The children's rights and sustainable development agendas in Wales, as at the international level, have emerged in parallel rather than symbiotically.

The first objective of this article is to review how these agendas have emerged and their impact on the exercise of governmental functions in practice. The extent to which these developments have opened pathways to fulfilling the requirement of legal enforceability of environmental, economic, social and cultural rights, which is implicit in the SDGs and explicit in the CRC, is also considered. The review embraced a detailed secondary based analysis of policy and legislation in Wales pertaining to sustainable development and children's rights between 1999 and 2020 and a consideration of international development discourse on both agendas during this time period.

The second objective of the article is to examine the importance attached to children's participation in decision-making relating to both agendas and some of the practices developed in Wales to support this. In this analysis, the authors applied a children's rights lens, aligned to Lundy's position that children's rights are not about pity, charity or (just) protection, or (just) the general principles of the CRC, and is not the same as 'well-being' or indeed the SDGs. Rather, it acknowledges children as legitimate and autonomous rights holders and recognises their agency and capacity for self-determination (Lundy 2019). As part of the examination of participative practices in Wales, this article features a discussion on the establishment of a National Youth Parliament (1999–2019) and the child led 'Lleisiau Bach Little Voices' projects (2014–2020). Both projects have contributed to local and national decision making on children's rights and sustainable development.

The third objective is to consider how the agenda on children's rights and the agenda on sustainable development have begun to develop in a more integrated manner in Wales, mainly due to the guidance provided by the Children's Commissioner for Wales and the Future Generations Commissioner to public bodies. The article concludes by explaining how this case study on Wales provides an example of how to build on a children's rights approach to sustainable development, while embedding participative practices and strengthening legal enforceability.

The findings of the case study are analysed and discussed in Sections 2–4, and concluding reflections are made in Section 5.

## 2. The Emergence of Two Separate Agendas in Wales

*2.1. Context*

Welsh devolution is both part of a process of significant constitutional change within the United Kingdom, beginning in 1997 and continuing at the time of writing, and a part of Wales' own particular history and characteristics (Jowell and O'Cinneade 2019). It also exemplifies what has been recognised as a global trend towards devolution of authority and resources from nation-states to regions and localities (Rodriguez-Pose and Gill 2003), a trend which brings opportunities as well as challenges for implementation of international obligations (For example, see Resolutions 296. 2010. REV and 365 (2014 and Recommendation 280 (2010) REV of the Congress of the Council of Europe, on best practices of implementation of human rights at local and regional level in member States of the Council of Europe and other countries).

Wales is a country within the United Kingdom, whose law and governance was absorbed by England in the 16th century (Watkin 2012; Gower 2013). Separate laws for Wales, on a small number of issues, began from the late 19th century, and during the 20th century, separate administration was established in fields including housing and education. In 1965 the Welsh Office was established as a UK Government Department of State. Previously, a national referendum on devolution for Wales in 1979 returned a negative response. However, in 1997 referenda in Wales, Scotland and Northern Ireland paved the way for devolution legislation conferring governmental powers on new institutions for each region. The Government of Wales Act 1998 established a single elected body, the National Assembly for Wales, with executive functions derived mainly from those exercised by the Welsh Office immediately prior to devolution. However, under the 1998 legislation, unlike Scotland and Northern Ireland, Wales had no separate parliamentary body. Successive further Acts of the UK Parliament have incrementally increased devolved governmental powers in Wales. This new devolved administration enjoyed the opportunity and the challenge to shape policies designed to meet the needs of the Welsh people. The National Assembly established in 1999 enjoyed executive powers to develop policy in the field of children's health, social care, education, transport, housing, play and leisure. The UK Government retained responsibility for the police and legal system, tax and benefits and policy relating to asylum, immigration and youth justice (Powers were transferred initially under the Government of Wales Act. 1998. Subsequent changes were made in the structure of Welsh devolution and the law-making powers of the devolved government and parliament by: the Government of Wales Act 2006; the Wales Act 2014 and the Wales Act 2017. An explanation of this progression can be found in the UK Government's *Devolution Guidance Note on Parliamentary and Assembly Legislation Affecting Wales*, available at the UK Government's official web pages: www.gov.uk, accessed 29 December 2020). The current position is that there is a Welsh Government and separate parliament, called initially the National Assembly for Wales and now, the Senedd. Wales remains, however, formally part of a single legal jurisdiction known as 'England and Wales' (Rawlings 2003).

Wales has a population of approximately 3 million people, with nearly 700,000 children under the age of 18 years (Office for National Statistics 2001). The geography of Wales is largely rural, with population concentrations in a small number of cities in the south, and with some areas of the country experiencing prolonged deprivation (Department for Work and Pensions 2013). Wales is a country of natural beauty and resources. It has three national parks, many Blue Flag beaches, and mountains which attract large numbers of tourists, bolstering the economy in the rural areas. Wales is officially a bilingual country (Welsh and English) and promotion of the Welsh language is both a policy priority of, and legal obligation on, the Welsh Government.

This case study begins at the turn of the century. In 1999, the first National Assembly for Wales was elected. At that time, both the Convention on the Rights of the Child and the Brundtland Report (World Commission on Environment and Development 1987) were just over 10 years old. In September 2000, world leaders came together at the United Nations Headquarters in New York to adopt the United Nations Millennium Declaration

(Office of the High Commissioner on Human Rights 2000). The Declaration committed nations to a new global partnership to reduce extreme poverty and set out a series of eight time-bound targets—with a deadline of 2015—known as the Millennium Development Goals (MDGs). This established a global partnership of countries where richer countries agreed to financially support poorer countries to achieve eight voluntary development goals and 21 targets for 2020 (UNDP n.d.b.). Human rights were reflected in the MDGs, although some commentators argued that they were not incorporated strongly enough (Williams 2013a). According to McInerney—Lankford, 'the relationship between human rights and development is arguably defined more by its distinctions and disconnects than by its points of convergence, despite substantial evidence of the potential for mutual reinforcement' (McInerney-Lankford 2009, p. 51). Alston referred to the two agendas as 'ships passing in the night' (Alston 2005, p. 755).

In Wales, meanwhile, sustainable development was stitched into the devolution settlement with a legal duty requiring the National Assembly for Wales to make a scheme setting out how it proposes, in the exercise of its functions, to promote sustainable development (Government of Wales Act 1998, sct.121). This set the foundation for experimentation and innovation on sustainable development (Davidson 2020). Children's rights, by contrast, emerged as a policy priority which would be woven into the fabric of Welsh devolved government by the new polity itself (Williams 2013b).

### 2.2. Children's Rights in Wales

Post-devolution, each of the devolved administrations was able to develop its own position in relation to the CRC, in addition to the national position taken by the UK government. In Wales, the political commitment of early devolved administrations (1999–2003; 2003–2007) to the CRC was striking (Clutton 2007; Williams 2007). The National Assembly for Wales enacted the first legislative reference to the CRC in the UK, when the Children's Commissioner for Wales, established in 2001, was to 'have regard to' the CRC in exercising their functions (Children's Commissioner for Wales Regulations 2001). In 2004 the National Assembly for Wales resolved that the CRC was its own overarching set of principles for all devolved policies on children and young people (National Assembly for Wales 2004). Key strategy documents, for example on partnerships for local child and youth services (National Assembly for Wales 2000), child poverty (Welsh Assembly Government 2005) and on young offenders (Welsh Assembly Government 2004b), made express reference to relevant articles of the CRC. Pressing for more powers was, in itself, a political imperative for the early Welsh devolved administrations (Morgan 2017). In the summer of 2009, the then First Minister, Rhodri Morgan, took advantage of the extension of legislative power permitted by the Government of Wales Act 2006 to announce his government's intention to 'explore further the possibility of introducing a Measure to embed the principles of the United Nations Convention on the Rights of the Child into law on behalf of Welsh children' (Williams 2013b). This was a landmark step in several ways. First, it attracted recognition for Welsh devolved law-making on the international stage, since the recommendation to incorporate the CRC into legislation by the Committee on the Rights of the Child in 2008 to the UK Government was taken on board in Wales (albeit within the limits of the devolved settlement). This was in contrast to the other devolved nations and the UK Government (Committee on the Rights of the Child 2008). Second, it further demonstrated the scope of policy differentiation, especially from England, made possible by Welsh devolution. Third, it gave legislative underpinning to a policy position on children's rights that had attracted a relatively high level of cross-party consensus in the National Assembly for Wales.

### 2.3. Sustainable Development in Wales

As noted above, Welsh devolution contained from the outset a requirement for a statutory scheme on sustainable development. The first such scheme, A Sustainable Wales: Learning to Live Differently, was drafted in a collaboration between environmental NGOs,

Welsh government agencies and the Sustainable Development Unit within the central Policy Unit of the National Assembly for Wales (Bishop and Flynn 2004). The overall strategic agenda set out by the Welsh Assembly Government in 2003 had a strong focus on the importance of sustainable development, stating that children and future generations should not be landed with a 'legacy of problems bequeathed by us' (Welsh Assembly Government 2003, p. 4). From this promising start, the major challenge became implementation and mainstreaming, stimulating a journey leading via scheme revision and development of indicators to the eventual enactment of the The Well-Being and Future Generations Act (Wales); (Davidson 2020).

As well as extending devolved law-making powers, the Government of Wales Act 2006 had made an important division between legislature and executive, which had been absent from the original scheme of Welsh devolution under the Government of Wales Act 1998. Now, the duty to promote sustainable development was placed on the executive rather than the legislative body, so the third sustainable development scheme, One Wales-One Planet, was published by the Welsh Government. It set out a primary goal of making sustainable development the central organising principle of the Welsh Government (Welsh Government 2009). This differentiated it from the earlier schemes. It used the following definition:

> In Wales, sustainable development means enhancing the economic, social and environmental wellbeing of people and communities, achieving a better quality of life for our own and future generations:
>
> In ways which promote social justice—and equality of opportunity and
>
> In ways which enhance the natural and cultural environment and respect its limits—using only our fair share of the earth's resources and sustaining our cultural legacy.
>
> Sustainable development is the process by which we reach the goal of sustainability.

This scheme was underpinned by two core principles: involvement of people and their communities; and integration (Welsh Government 2009, p. 26). These were to be universally applied in all decision-making (Senedd Research Note: Sustainable Development Structures and Policy 2015). The core principles were supported by six additional principles to be applied as relevant to the context: the precautionary principle; the polluter pays principle; the proximity principle; reducing our ecological footprint; taking account of full costs and benefits; and reflecting distinctiveness (Welsh Government 2009, p. 26).

These were further supported by 44 indicators, with a requirement to report against them annually using a traffic light system (Senedd Research Note: Sustainable Development Structures and Policy 2015). In 2011, Jane Davidson, a highly influential advocate for the environment and the then Minister for Environment, Sustainability and Housing, secured a further commitment in the Welsh Labour Party Manifesto to 'embedding sustainable development as the central organising principle in all our actions across government and all bodies', and to put this into legislation (Welsh Labour 2011, p. 92). Additionally, in 2011, following the closure of the UK Sustainable Development Commission, the Welsh Government appointed Wales' first Sustainable Futures Commissioner. No such equivalent was set up in any of the other three nations.

### 2.4. Children's Participation in Decision Making

A common, if not identical, thread in Welsh post-devolution policy on both children's rights and sustainable development was support for children to contribute to decision-making. In the early years of devolution, Wales was quick to develop formal mechanisms including legal requirements for schools to have pupil councils (School Council (Wales)), and for local authorities to implement structures for youth participation in decision-making (Children and Families (Wales)). Welsh Government also funded the charity Keep Wales Tidy to support Eco-Schools, a global programme aimed at empowering children and young people to make positive environmental changes to their school and wider community.

One of the most radical initiatives was the development of a Children and Young People's Assembly for Wales, the result of two earlier initiatives which eventually came together through support given by the National Assembly for Wales to a charitable company called Funky Dragon. Funky Dragon supported a Grand Council of initially 66 and eventually 100 young people elected by their peers in local authorities or representing special interests or minorities from the 22 Welsh local authorities (Croke and Williams 2018). The principal aim of Funky Dragon was to give young people 0–25 years old the opportunity to exercise their right to be heard. Ministerial oversight for the body was delivered by the then Minister for Education and Lifelong Learning, Jane Davidson.

In the early years of Welsh devolution, the Welsh Government made notable efforts to engage young people in sustainable development in Wales (Welsh Assembly Government 2004a). This may have been primarily because the then Minister for Education and Lifelong Learning, Jane Davidson was, as earlier mentioned, also a powerful advocate for the environment. Ten young people attended the United Nations World Summit on Sustainable Development in Johannesburg in 2002, with Rhodri Morgan, the then First Minister, presiding over talks with 20 regional and sub-national governments. This resulted in an agreement being reached to set up a global network of countries to work together on sustainable development (Netherwood and Flyn 2012). The First Minister undertook to involve young people through both Funky Dragon and the Wales Youth Forum on Sustainable Development. This then led to young people from both these forums taking part in a Conference in Cardiff on Sustainable Development for regional governments from around the world in 2004 (Welsh Assembly Government 2004a). The Welsh Government was also committed to ensuring that the principles of sustainable development were included in schools and further and higher education institutions. By 2006 Welsh Government had published a strategy that advised how Education for Sustainable Development and Global Citizenship should be implemented across all sectors of education (Welsh Assembly Government 2006).

From 2006, Funky Dragon's young Grand Council began to carry out research and produce reports on their experience of achieving their rights and a steering group of young people took their recommendations to key decision makers, including Welsh Government Cabinet and the UN Committee on the Rights of the Child in 2008 (Croke and Williams 2018). The 'Our Rights: Our Story' 2008 research engaged widely with young people (11–18 years) in Wales, collecting some 10,035 survey responses, holding 140 interactive workshops and conducting 37 interviews with young people from specific interest groups (Funky Dragon 2008).

This process produced 66 recommendations covering broadly, themes of Education, Information, Health and Participation (Croke and Williams 2018). Interestingly, there was not a clear focus on 'sustainable development' in this youth-led initiative. However, themes selected by the young researchers spanned economic, cultural, societal and environmental considerations relevant to sustainable development, including a concern for children in the future as well as the present (Funky Dragon 2008). Funky Dragon was cited as an example of good practice by both the Committee on the Rights of the Child and in the international guidance to NGOs on how to involve children and young people in reporting to the Committee on the Rights of the Child (NGO Group for the CRC 2011).

The involvement of young people (11 years+) in this UN reporting process brought to attention the under-representation of younger children. The steering group were keen to address this and carried out smaller scale research activities with 2,525 children, presented in a final report, 'Why do people's ages go up not down?' Funky Dragon developed the 'children as researchers' methods later deployed in 'Lleisiau Bach Little Voices' projects as a means to allow younger children (typically 7–11 years old) to identify and prioritise their own areas of concern, but to also have autonomy over the way in which they were represented and presented (Dale and Roberts 2017).

This first 10 years of devolution demonstrated that Wales was strongly committed to embedding sustainable development and children's rights into both law and policy. These

were divergent as well as innovative developments in the context of the UK. However, the two agendas did not converge nationally in Wales. Wales also developed innovatory approaches to support children's right to be heard, through national and local mechanisms contributing to children's rights and sustainable development. However, the two agendas—children's rights and sustainable development—were led and supported separately in devolved Welsh government, rather than being integrated together.

## 3. The Development of Two Pieces of Radical Legislation in Wales

### 3.1. Context

Globally by 2011, the children's rights-focused agenda had become increasingly supported by international NGOs, such as Save the Children International (Save the Children Sweden 2005) and the UN agency for children, UNICEF (Vandenhole 2014). The importance of human rights had become more integral to all UN agencies (Office of the High Commissioner on Human Rights 2012). The UNDP recognised the importance of human rights to its development work, and a major outcome of the Rio +20 conference in 2012 was the commitment to establish a suite of Sustainable Development Goals (SDGs) (Senedd Research Note: Sustainable Development Structures and Policy 2015). 17 SDGs were published in 2015 by the UNDP; they included: poverty eradication, action on climate change, access to education, gender equality, action to halt biodiversity loss, economic growth and sustainable energy. These interconnected goals clearly impact on children and are part of a call to action for Agenda 2030 (UN n.d.).

Nolan comments that this Agenda 2030 addresses the human rights blindness of the Millennium goals and includes extensive reference to human rights grounded in the Universal Declaration of Human Rights. She also acknowledges their strong reference to issues that impact on children and as a vehicle for galvanising resources and institutional support to children. However, she critiques them for ultimately being policy goals and not legally binding obligations, with limited processes of monitoring and accountability. She comments that of great concern is that states will defer to the SDGs and will try to rid themselves of the burden of obligation and accountability, therefore undermining children's rights (Nolan 2019).

Kilkelly also expresses disappointment that the Goals are not articulated as legal rights and that they hardly mention the CRC. Even though UNICEF produced a document that links the Goals in a complex matrix to the CRC (UNICEF 2020), the Goals themselves are not expressed as human rights and thus can only be seen to be political claims or charitable requests, not legally enforceable entitlements (Kilkelly 2020). Nolan additionally forcefully comments that the SDGs are more consistent with top-down approaches to development and do not recognise children as agents of change and rights holders, which is instrumental to the CRC (Nolan 2019).

However, the CRC itself can also be critiqued. The CRC does make reference to the right to a healthy environment within Article 24, and General Comment No. 15 emphasises that climate change is the 'biggest threat to children's health and exacerbates health disparities' (Committee on the Rights of the Child 2013a, para 50). However, apart from asking that States put child health at the centre of their climate change strategies, there is no further reference to climate change. As the Office of the High Commissioner on Human Rights acknowledges, climate change has a disproportionate impact on children (Office of the High Commissioner 2016). Arts comments on the striking lack of reference to climate change throughout the Concluding Observations and General Comments (Arts 2019). The intergenerational equity principle is also absent (Atapattu 2019). Arguably a General Comment focused on sustainable development, including climate change, environmental impacts and future generations should be developed with urgency, to aid interpretation of the relevant articles of the CRC.

Turning to the Welsh context, the second decade of devolution saw the introduction and passing of radical legislation to further embed children's rights and sustainable development into the Welsh legislative framework. However, the two agendas remained

disconnected. The next section discusses their strengths and weaknesses and further explains how Welsh Government support for a national youth platform was disrupted although various non-governmental initiatives continued to develop practices of empowerment for children.

### 3.2. Children's Rights

In 2011 in Wales, the Rights of Children and Young Persons (Wales) ('the Rights Measure') was a stand-out representation of Wales' capacity for innovative, child rights-focused, internationally informed law-making. Within the UK, it was the first general legislative provision about CRC implementation, an achievement in which the National Assembly for Wales could take pride, especially given that Wales still at that time had the weakest legislative competence amongst the devolved governments and parliaments within the UK.

The degree of innovation, at this time, should not be exaggerated. The central mechanism of the Rights Measure was borrowed from UK legislation on equalities enacted shortly before the Welsh Measure was published (Equality Act 2010 s. 149: the 'public sector equality duty'). It was a duty to have 'due regard' to the CRC placed upon Welsh Government Ministers, rather than a requirement to comply with the requirements of the CRC (Section 1 Rights of Children and Young Persons (Wales)). This stands in contrast to Sections 6 and 7 Human Rights Act 1998 (Human Rights Act 1998) which make unlawful acts by public authorities which are incompatible with the European Convention on Human Rights and confer an individual right of legal action on a victim of a violation. In the Rights Measure, enforcement is by political and administrative scrutiny, with judicial review as a backstop. There is no individual remedy for a rights violation (Hoffman and Williams 2013).

A key implementation tool under the Rights Measure is a child rights impact assessment (CRIA) (Committee on the Rights of the Child 2013b). This is not required explicitly, but the Rights Measure does require the Welsh Ministers to periodically review, and to report on, a scheme setting out how they are fulfilling their duty of 'due regard' (Rights of Children and Young Persons (Wales)). Within the statutory scheme, an approach to child rights impact assessment has evolved. There is a template and guidance for policy officials to follow. Some 260 CRIAs were carried out between 2012 and 2018 (Hoffman 2020). An evaluation published in 2015 concluded that it had embedded the core objectives of human rights impact assessment into government decision-making, but that implementation was often inconsistent and a departure from good practice because of a failure to consult with children, implementation late in the policy process, and limited knowledge of the CRC (Hoffman 2020). A study of legal integration of the CRC in Wales in 2018 largely reiterated these findings about CRIA (Hoffman 2020) and was further reinforced by the Welsh Parliament's (now called the Senedd) Children, Young People and Education Committee in 2019. After a national inquiry into children's rights in 2019, this Senedd Committee published a strong set of recommendations; including that CRIAs must be published and made available and that a decision not to do so should be challengeable, and also that the duty of due regard should be extended to all public bodies (Senedd Children and Young People and Education Committee 2020).

The inquiry revealed that the reality for many children, was that their rights were still not being realised and that a stronger framework of governmental accountability was required, with legal enforceability of children's rights being strengthened (Senedd Children and Young People and Education Committee 2020). The majority of recommendations were accepted by the Welsh Government. However, the strengthening of legal enforceability was not. Children's sector organisations, who gave evidence as part of the inquiry, pushed back further requesting that the Senedd consider tabling legislation that considers extending the duty of due regard and also adopting a compliance duty (Children's Commissioner for Wales 2020; Children in Wales 2020; Connor et al. 2020).

A duty of due regard on public bodies would mean that like the duty placed on Welsh Government to have due regard to the CRC, in the exercise of all their functions under the

Rights Measure, all public bodies would also have to adhere to this requirement. With the introduction of a compliance duty into legislation, this would mean that all public bodies in Wales would have to comply with the principles and provisions of the CRC and children would have the right to seek a remedy for a rights violation in a court of law. This would be similar to the duty to act compatibly with the ECHR (Section 3 (1) Human Rights Act 1998) in the UK Human Rights Act 1998). Children's sector organisations believe that public bodies will more likely act in compliance with their duties under the CRC if they know that children whose rights are breached by lack of compliance to the CRC can bring an action for a remedy. Kilkelly explains how 'relatively weak measures of implementation can help to build momentum in favour of stronger measures' as seen from the recent experiences of Scotland and Sweden who have moved towards a more direct incorporation of the CRC (Kilkelly 2019, p. 327). Hence in Wales, moving from a duty of due regard to a compliance model would create opportunities for greater accountability whilst continuing to promote awareness of children's rights amongst public bodies and decision-makers. The Scottish Government, one of the other devolved nations, have agreed to this step in the United Nations Convention on the Rights of the Child (Incorporation) (Scotland) Bill). The Welsh Government, however, did not move its position in a plenary debate that took place in early 2021 (Senedd 2021).

### 3.3. Sustainable Development

Returning to the policy agenda for sustainable development, between 2012 and 2014, the Sustainable Futures Commissioner led a national conversation with over 7000 people, entitled The Wales We Want, to challenge people to consider issues beyond the constraints of their daily lives (Commissioner for Sustainable Futures 2015). Wallace notes that the complexity of communication hindered putting sustainable development at the heart of policy making (Wallace 2019). This led to the re-framing of sustainable development in Wales within a broader concept of 'well-being', the thinking being that sustainable development was more narrowly understood by stakeholders as a concept related only to environmental concerns (Wallace 2019). After the Welsh Government consulted on its Sustainable Development Bill White Paper in 2012–2013, it laid the ground-breaking Well-being of Future Generations (Wales) Bill on 7 July 2014. This was to be a global first: legislation that was designed to safeguard the well-being of future generations (Davidson 2020).

In 2015, the Well-being of Future Generations (Wales) Act 2015 ('WFGA 2015') was passed. The Act places a duty on all public bodies to carry out sustainable development and improve and achieve economic, social, cultural and environmental well-being. All public bodies must ensure that when making decisions they take into account the impact they could have on people living in Wales in the future. The Act states that:

> Each public body must carry out sustainable development. The action a public body takes in carrying out sustainable development must include: a. setting and publishing objectives ('well-being objectives') that are designed to maximise its contribution to achieving each of the well-being goals, and b. taking all reasonable steps (in exercising its functions) to meet those objectives.

The seven well-being goals are listed and described as follows (Table 1).

**Table 1.** Seven Well-Being Goals in WFGA 2015 (The Well-Being and Future Generations Act (Wales)) Part 2, Section 4).

| Goal | Description of the Goal |
|---|---|
| A prosperous Wales | An innovative, productive and low carbon society which recognises the limits of the global environment and therefore uses resources efficiently and proportionately (including acting on climate change); and which develops a skilled and well-educated population in an economy which generates wealth and provides employment opportunities, allowing people to take advantage of the wealth generated through securing decent work. |
| A resilient Wales | A nation which maintains and enhances a biodiverse natural environment with healthy functioning ecosystems that support social, economic and ecological resilience and the capacity to adapt to change (for example climate change). |
| A healthier Wales | A society in which people's physical and mental well-being is maximised and in which choices and behaviours that benefit future health are understood. |
| A more equal Wales | A society that enables people to fulfil their potential no matter what their background or circumstances (including their socio economic background and circumstances). |
| A Wales of cohesive communities | Attractive, viable, safe and well-connected communities. |
| A Wales of vibrant culture and thriving Welsh language | A society that promotes and protects culture, heritage and the Welsh language, and which encourages people to participate in the arts, and sports and recreation. |
| A globally responsible Wales | A nation which, when doing anything to improve the economic, social, environmental and cultural well-being of Wales, takes account of whether doing such a thing may make a positive contribution to global well-being. |

The Act links the duty on sustainable development to public sector reform, prevention, collaboration, integration, involvement and long-term thinking, described as five ways of working (see Section 5 of the Act) (The Well-Being and Future Generations Act (Wales)) sct. 5). The duty is placed on 44 public bodies and they must demonstrate how they have applied the sustainable development principle in practice (Auditor General for Wales 2017). Public Service Boards (PSBs) were established for the integrated planning of service delivery and the development of well-being assessments (Senedd Research Note: Sustainable Development Structures and Policy 2015). A Future Generations Commissioner was created as part of the legislation to promote the principles of the Act and to support public bodies to implement the legislation ((The Well-Being and Future Generations Act (Wales)), part 3). Working alongside the Auditor General for Wales, this statutory body is also responsible for monitoring the Act (The Well-Being and Future Generations Act (Wales)), sct. 15).

Wallace argues that the Act places an equal weight on all aspects of well-being, challenging the traditional dominance of economic over societal, environmental and cultural considerations (Wallace 2019). However, a lack of focus on the environmental pillar was initially subjected to criticism during the passage of the bill by the environmental advocates who argued that environmental concerns were not sufficiently addressed (National Assembly for Wales Environmental and Sustainability Committee 2014).

At the same time, human rights advocates argued that the focus on well-being had the effect of suppressing rights. The Wales UNCRC Monitoring Group claimed that the Act:

> does not give sufficient focus to the enforcement of human rights which we believe is a precondition for sustainable development and a prosperous Wales. Without acknowledging and acting to realise the human rights of people, sustainable development is not possible. We believe that the delivery of public services in Wales must be done through a human rights lens and that the Future Generations Bill presents us with a key opportunity for a human rights framework to be enshrined into law. (Wales UNCRC Monitoring Group 2014)

This is supported by research by Bangor University School of Law, in which some research participants felt that the focus on well-being in the legislation 'as the cornerstone to

good administration has led to the marginalisation of other foundations, e.g., human rights, equality, and specifically, principles of administrative justice' that had been developed for Wales . . . ' (Bangor University School of Law 2020, p. 1). These criticisms resonate with the missed opportunity referred to by Kilkelly in relation to the development of the international SDGs (Kilkelly 2020). Legislation framed in terms of public services achieving well-being objectives instead of enforceable human rights, risks having the effect of weakening rights protection.

As Wallace comments, the Act stops short of requiring the objectives to be met (Wallace 2019). Lord Thomas goes as far to say that aspirational legislation, including WFGA, raises false hopes and undermines the rule of law (Lord Thomas of Cwmgiedd 2019). The WFGA does not give the right to an individual or a group of individuals to bring a claim for judicial review based on an allegation of a breach of the act, giving rise to criticism that it is 'toothless' (Martin 2019).

Additionally, the Commission on Justice in Wales noted that the 'far-sighted policies on sustainability, future generations and international standards on human rights are not integrated into the justice system' (Commission on Justice 2019, p. 456). There is also a lack of capacity at local authority level with officials struggling to implement often perceived, complex legislation without sufficient resources or support (Bangor University School of Law 2020) as well as a plethora of plans and competing strategies (Wallace 2019). This was recognised also at the international level by UN Rapporteur for Health Paul Hunt (2002–2008) who commented that practitioners need more practical guidance to translate complex legal provisions into practice (Hunt 2016).

### 3.4. Children's Participation in Decision Making

#### 3.4.1. A Welsh Youth Parliament

Doel-Mackaway explains that participation of children is a critical component of democracy, good governance and promotion of the rule of law and is therefore essential for sustainable development (Doel-Mackaway 2019). After the earlier cited achievements of securing a national democratically elected platform for young people, it may come as a surprise that Funky Dragon came to an abrupt end when they ceased to be funded by the Welsh Government in 2014, just a year before the passing of the WFGA 2015. The circumstances leading to this cessation in funding are discussed elsewhere (Croke and Williams 2018). Suffice to say, that with hindsight it seems clear that tensions arose in part because, following the separation of powers under the Government of Wales Act 2006, funding became a function of the Welsh Government (the executive) and not the National Assembly for Wales (the parliament), and this was exacerbated by the onset of austerity measures following the global financial crisis of 2008 (Croke and Williams 2018). The demise of Funky Dragon left many of its former members, staff and supporters bewildered and angry, especially because the young people had not been consulted about the proposals (Croke and Williams 2018). The Welsh Government had not met the requirements of adhering to the child's right to be heard under Article 12 of the CRC, as laid out in the Rights Measure (Wales UNCRC Monitoring Group 2014).

Between 2014 and 2018, Wales was in the regressive position of not having a national democratic platform for children. However, a campaign was launched by the remaining voluntary trustees of Funky Dragon, encompassing commissioned comparative research, a public consultation and deployment of the dynamic of the CRC monitoring process to press the case (Croke and Williams 2018; Woll 2000). It was ultimately endorsed by the Presiding Officer of the National Assembly for Wales who then set in train a process leading to the establishment, in its proper constitutional place, of the Welsh Youth Parliament (National Assembly for Wales 2016). The frustration felt by young people who had become accustomed to the idea of a national youth parliament was encapsulated by one of Funky Dragon's young trustees:

> We constantly hear in the press about disengaged and politically disaffected young people, but young people are not asked for their views on the EU ref-

erendum and the perception instead is that young people are more engaged with football than politics and to top it off one of the headline stories is about 'Voles' . . . . . . nobody has reported on our positive news story about a group of committed young people who are campaigning for a democratically elected voice for the young people of Wales and desperately want all young people to have the opportunity to engage and influence local, national and global politics. (Croke and Williams 2018, p. 17)

In 2016, even with the Rights Measure and the recently passed WFGA 2015, children across Wales were not consulted on what was to have a significant impact on them and future generations; the decision to leave the European Union. This further exposed the weaknesses in public bodies' understanding of the well-being of future generations and indeed the Welsh Government's obligation to have due regard to children's right to be heard in all matters that affect them.

It is equally striking, however, that the campaign for the Welsh Youth Parliament found support in all political parties represented in the National Assembly for Wales. On Saturday 23 February 2019, the Welsh Youth Parliament met for the first time. It is too early to evaluate the success of this initiative, but this national democratically elected body of young people is now firmly situated on the parliamentary side of governance and can thus openly criticise and better hold the executive to account. Funky Dragon was a charitable organisation which was funded and overseen by the executive from 2006, which made it vulnerable to short-term funding, particularly as the young people were directly criticising the executive in their reporting to the UN Committee on the Rights of the Child (Croke and Williams 2018). The new youth parliament is free to voice their opinions and promote the realisation of children's rights and the well-being of future generations.

### 3.4.2. Lleisiau Bach Little Voices (LBLV)

While Funky Dragon ceased operations in 2014, the Lleisiau Bach Little Voices (LBLV) 'children as researchers' had just begun a new project working with primary schools and communities in Wales, with funding from the National Lottery Community Fund. The Observatory on Human Rights of Children at Swansea and Bangor Universities took over management of the project. During this phase of the project, a milestone was the submission of a child-led, under-11 year olds' report to the UN Committee on the Rights of the Child in 2015. This innovative project empowered younger children to speak out and influence local and national decision making on issues that covered all the pillars of sustainable development: environmental, cultural, social and economic concerns. This section focuses on LBLV's children as researchers approach that empowered children to be researchers and activists, inspiring younger children to become change agents, locally and nationally.

The methodology deployed by LBLV is a form of participatory action research (PAR) as defined by Hart who described as its main features:

> that the research be carried out by or with the people concerned; the researcher feels a commitment to the people and to their control of the analysis; research begins with a concrete problem identified by the participants themselves; and it proceeds to investigate the underlying causes of the problem so that the participants can themselves go about addressing these causes. (Hart 1992, p. 16)

However, the LBLV methodology has two features that distinguish it from most if not all other PAR practices. First, it is framed by the CRC—as to the scope of projects, conduct of research, ethics and approach to impact; and second, the children select and own the research project. The LBLV project worker provides support and coordination (Dale and Roberts 2017).

The approach echoes Larkin et al. who argue that for the children's rights research community to have a strong contribution to achieve influence through their actions, there is an imperative need to start from children's self-identified concerns and then identify the

relevant range of moral, legal, political or economic rights that may provide resources for their activism (Larkins et al. 2015). Quennerstedt comments that:

> Instead of prioritising the universal and a top-down approach in research, where the urgent research questions spring from universal claims, the opposite position is taken, priority is given to context, particularity and a bottom-up approach ( Quennerstedt 2013, p. 244)

Additionally, in academic research, both the substantive and procedural requirements of the CRC Convention have stimulated increased use of methods which involve children as active participants or co-producers (Boyden and Ennew 1997; Barnen and Kirby 2002). There is debate about whether academic research 'on' children should now normally be done 'with' or even 'by' children (Chae-Young et al. 2017). Researching 'with' or 'by' rather than 'on' children is seen as supportive of implementation of children's rights (Beazley et al. 2009; Kellett 2010; Lundy et al. 2011).

Thanks to continued support from the universities and the National Lottery Community Fund, by 2020, the LBLV team was providing: flexible, direct or indirect support for children in their own localities to do their own research, build their case for change and make change happen; two-way connectivity with impact partners and decision-making processes, whether at community, local, national or international level; and training and mentoring for professionals to use the approach in their own work. By 2020 the LBLV team had conducted some 120 local projects along with national child-led surveys and engagements feeding into various decision-making processes. The team had worked with over 1000 children as researchers, and many thousands more children, had been involved as project participants using methods like surveys, research days, photo-voice and interviews. Many adults were also involved, whether as gatekeepers, helpers, impact partners or audience for the children's research-based recommendations. Selection of child participants was negotiated with gatekeepers, prioritising, where practicable, children who were not otherwise engaged in participative activities such as school councils or eco-schools committees (Dale and Roberts 2017). Research with children was conducted in both the English and Welsh languages.

The work had evolved from human rights monitoring under the CRC, and the team devised a taxonomy derived from the reporting categories adopted by the Committee on the Rights of the Child to help distil from the projects an appreciation of the issues and concerns that children wanted to explore and things they wanted to change. The aim of this was to enable key messages from the children's cumulative efforts to be fed back to the Committee responsible for monitoring implementation of the Convention (Little Voices Shouting Out 2015).

The themes are:

A. Disability, basic health and welfare (includes health, health promotion, well-being)
B. Education/leisure/cultural activities (includes school, training, skills and employability)
C. Environment and amenities (includes climate action, pollution, conservation, circular economy)
D. Road Safety
E. Food
F. Play, leisure, recreation, culture and art
G. Community (includes topics relating to general public well-being, poverty, social exclusion affecting others, public security and public services)
H. Civil rights and freedoms (includes identity, information, privacy/image, freedom of association, assembly, thought, conscience and religion, non-discrimination and remedies)
I. Knowledge of rights/CRC
J. Special protection

Where topics identified by children could be related to climate action or environmental protection, in the above classification they are treated as part of 'Environment and Amenities'. This category would also include local environmental issues which could be about school grounds or a local park in the children's immediate environment. Litter, dog mess and, increasingly, plastic waste and recycling were recurrent issues.

Thematic analysis of the LBLV project data shows children selecting as potential or actual research topics, issues about deforestation, plastics pollution, re-cycling and wildlife preservation, alongside a remarkably wide range of other topics. Some local projects produced relevant recommendations and impacts. For example, a primary school in south Wales deployed the methodology effectively to engage community leaders, local businesses and others in a plastic waste collection project.

By 2019, environmental issues had become sufficiently popular with LBLV groups for the team to be able to convene 4 'climate action summits' at which project groups could exchange knowledge and experiences, the last of which was combined with collection of children's views to feed into a Welsh Government consultation on Circular Economy policy (Welsh Government Circular Economy 2020).

The LBLV work is cited in Jane Davidson's book #futuregen (Davidson 2020). Understandably, she foregrounds (and elaborates upon) the environmental protection aspects of the work:

> What is particularly interesting . . . is that primary-school-aged children through their research are calling for action on the key issues of the day: deforestation, habitats for wildlife, plastics in the ocean, endangered animals and the impact of fossil fuels.

> In Wales they want more trees, more locally grown food, more wildlife. They want to see electric/hydrogen cars, less plastic, fewer factories, reduced carbon emissions and more recycling and reusing . . . a fairer and more tolerant society which provides better support for those who are homeless or in poverty; is less selfish, kinder, more accepting of others; and most importantly listens to the views of children and young people. (Davidson 2020, p. 123)

Certainly, the project data supports a view of children—especially those under 11—identifying, initiating or connecting with the kinds of transformational initiatives that we know are necessary to achieve the SDGs or, in #futuregen-speak, 'well-being'. It is important to note however that majority of the children's projects' recommendations are not about action to be taken at macro or exo-system levels. They focus rather on what the children themselves can do in partnership with others in their immediate communities: growth and production of own food produce at home and at school, purchasing of locally sourced food, tackling food waste, the excessive use of single use plastics, the call for greener school spaces and free solar panels for schools: in other words, the sphere in which they can bring about change rather than the sphere in which they cannot.

In the wider context they identify broader action to be taken in partnership with others who possess decision making powers to achieve the SDGs: making public transport cheaper, increased access to affordable electric cars, tax on plastic packaged items, the use of natural products in clothes and widespread schemes that ensure supermarkets make use of their food waste by donating to foodbanks and homeless shelters. The LBLV projects did not constrain children to policy siloes or adult-led agendas. Instead, they empowered children to identify their own issues of concern in both their localities and the wider world, and to challenge and transform the world around them.

The LBLV team also helped the Campaign for a Children and Young People's Assembly for Wales to bring about the Welsh Youth Parliament and mentored the first Welsh Youth Parliamentarians in building their evidence-based case for change in their priority areas, which were: Life Skills in the Curriculum; Child and Adolescent Mental Health; and Plastic Waste. The LBLV team also supported children to feed into the Senedd Children, Young People and Education Committee 2020 national inquiry into children's rights (Senedd Children and Young People and Education Committee 2020). The experience of LBLV

and the campaign for a youth parliament in Wales demonstrates that the CRC can be used to produce a dynamic for change, both at national level and in immediate, everyday environments (Croke and Williams 2018).

The second decade of devolution in Wales brought forward radical legislation on both children's rights and sustainable development. However, early indications question how effectively they were working and whether stronger legislation that supported firmly entrenched social, economic, cultural and environmental rights would do better to ensure that public bodies are held to account. The initial demise of a national democratic platform for young people was met with considerable concern, but non-governmental actors, using the dynamic of the CRC and a combination of charitable and institutional resource, helped to continue support for children to speak out and influence change.

## 4. The Integration of Children's Rights and Sustainable Development

Vandenhole has described how a children's rights approach to sustainable development is still very much under construction (Vandenhole 2019). However, in Wales, shortly after the introduction of the SDGs, 'children's rights were re-imagined through the prism of sustainable development' and vice versa (Davies 2019, p. 33). This initiative was begun when Hoffman and Croke were commissioned to produce a children's rights approach (CRA) statement and guide for public bodies for the Children's Commissioner for Wales (Hoffman and Croke 2016). The reason for the development of this guide was that consecutive reports to the Committee on the Rights of the Child had reported on a policy to implementation gap (Croke and Williams 2015; Croke 2013; Croke and Anne 2007; Croke and Anne 2006). Development of successful policy on children's rights at the national level was identified as failing to be translated into practice by public bodies. The Guide was developed as a principled and practical framework to give advice to public bodies regarding how to adopt a children's rights approach to their work. Hoffman and Croke identified five principles (Hoffman and Croke 2016) the principles are: Embedding children's human rights, Equality and Non-discrimination towards children, Accountability to children, Participation of children, and Empowering children. These principles bear similarities to the principles included in other human rights approaches and children's rights approaches (Lang et al. 2011) and were identified as being the principles best suited to drive the implementation of a CRA for public bodies in Wales. The Guide is now widely adopted in Wales as the framework for a CRA for public services and the Children's Commissioner for Wales has named it 'The Right Way' (Children's Commissioner for Wales 2017).

This framework was also utilised in the development of guidance issued by the Children's Commissioner for Wales and the Future Generations Commissioner in 2017, to develop practical tools and examples to help public bodies consider children's rights across each of the Well-being goals and the Five Ways of Working under the Well-Being of Future Generations Act (Children's Commissioner for Wales and Future Generations Commissioner 2017). The Commissioners state:

> As Wales' independent commissioners for Children and Future Generations we have distinct roles but common interests. We want to enable public bodies to put children's rights to be safe, healthy and to flourish here and now at the centre of their planning and delivery. We also want to ensure that they plan for the long-term—for the rest of the lives of children living in their communities now, and for future generations still unborn. We have worked together to consider how Wales' commitment to internationally recognised children's rights can work with the groundbreaking Well-being of Future Generations Act to meet children's needs—now and in the future. (Children's Commissioner for Wales and Future Generations Commissioner 2017, p. 5)

> This is a significant effort and possibly also a global first, to support public body officials with practical tools to understand how the two agendas complement and reinforce each other and how they should be implemented (Figure 1). It demonstrates how the CRC can contribute to the sustainable development principle

and considers how the five principles of a CRA can be embedded across the five ways of working of the WFGA 2015. It offers case studies to help the 44 public bodies understand how the 5 principles of a CRA can be translated into practice. It suggests that to demonstrate how Public Service Boards (PSB) have paid due regard to the CRC when developing their Well-being Objectives and Well-being Plans a children's rights impact assessment should be conducted. The Children's Commissioner for Wales has developed a standard Children's Rights Impact Assessment for all public bodies to use and a simple self-assessment tool kit for services to measure the extent to which services give children and young people access to their rights, and how to plan improvements (Self Assessment Tool Kit and Children's Rights Impact Assessment n.d.). This responds in part to Nolan's critique of the international SDGs, by not neglecting children's rights monitoring mechanisms. (Nolan 2019)

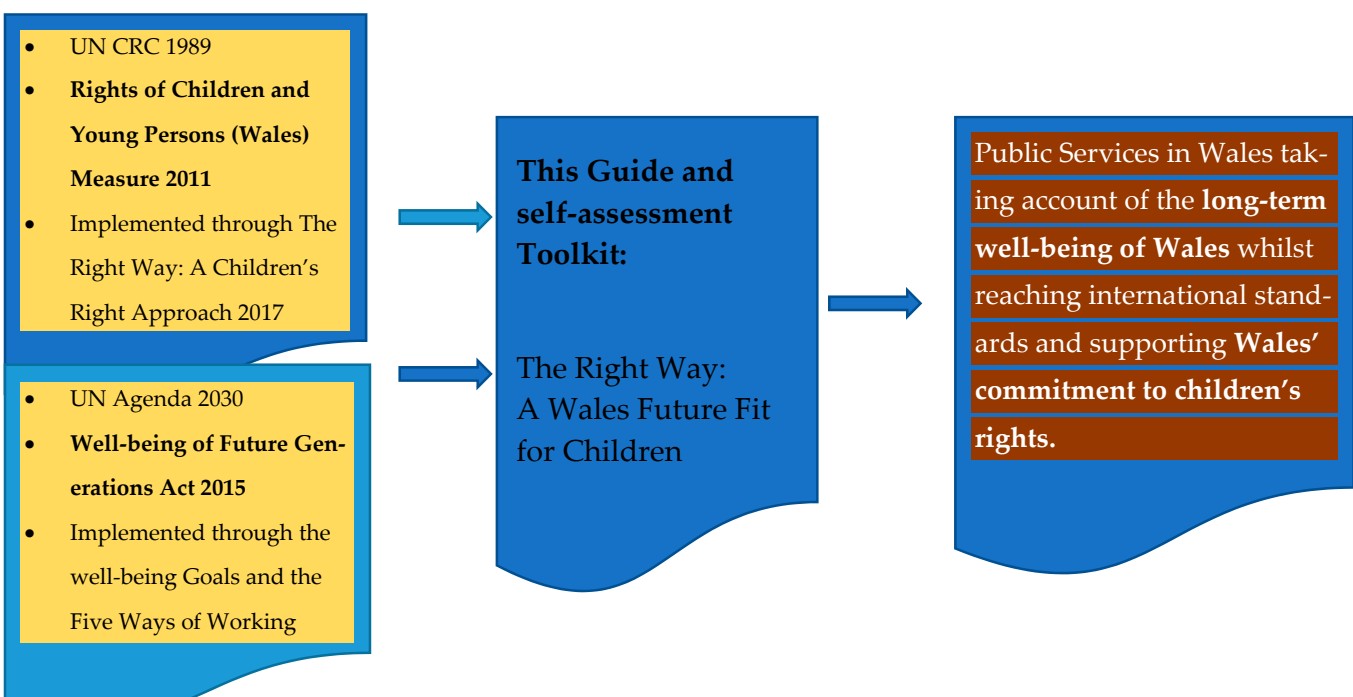

**Figure 1.** The Right Way: A Wales Future Fit for Children (Children's Commissioner for Wales and Future Generations Commissioner 2017, p. 5).

Critical to the combined approach, is an understanding of prioritising children, and giving them a seat at the table when a PSB has to consider the implementation of the Well-being objectives. The 5 principles of the CRA have contributed to a conceptualisation that envisions children as rights holders, with the capabilities to take advantage of their human rights, to advocate for the human rights of others as well as future generations. It challenges public bodies to be accountable to children currently and to consider future generations. This also challenges concerns recognised by Desmet in her evaluation of international environmental and sustainable development policy that the 'potential for recognising the rights and agency of children remains to a certain extent untapped' (Desmet 2019, p. 215).

Linked to the agenda on the realisation of children's rights, the Future Generations Commissioner has also identified work on Adverse Childhood Experiences (ACE) as a priority for action (Adverse Childhood Experiences in Relation to the Work of the Future Generations Commissioner 2020). The aim has been to focus on improving the well-being of future generations by preventing harm in early childhood. Wallace reports that 16 out of the 19 Public Service Boards had identified ACEs as one of their priorities (Wallace

2019). This further evidences a focus on children. Throughout a Wales Future Fit for Children, best practice case studies are included that demonstrate how public bodies are already implementing a children's rights approach. For example, Pembrokeshire County Council made a corporate commitment to children's rights which has been translated into: strengthening their representative bodies; supporting and improving school councils; increasing adults' awareness of children's rights; and promoting the active citizenship of children and young people. They have also empowered children to assess corporate planning against standards of the UNCRC. The City and County of Swansea has recognized the importance of embedding children's rights into workforce development planning, and this has resulted in mandatory children's rights training for staff, children's rights being included in their Strategic Equality Plan and children being supported by staff to influence budgetary decisions. Abertawe Bro Morgannwg Health Board has established a Children's Rights Charter and a young people's advisory group that influences health services delivery and decision-making across the health authority. Rhondda Cynon Taff Local Authority have adopted the UNICEF Rights Respecting Schools initiative for all their secondary schools, encouraging whole school approaches to children's rights; developing rights literacy of children and teachers. These are just some examples that demonstrate how some public bodies in Wales are aiming to embed a children's rights approach that contributes to realising children's rights and the objectives of the WFGA 2015 (The practice examples are drawn from (Children's Commissioner for Wales and Future Generations Commissioner 2017).

## 5. Concluding Reflections

Wales, despite being a devolved nation with limited, albeit increasing powers, has taken bold steps towards embedding children's rights and SDGs reform into law. From examining the issues that have arisen in the Welsh context and the ways in which they are beginning to be addressed, a pathway can be discerned towards better integration of the two agendas. This pathway consists of building on a children's rights approach to sustainable development, embedding participative practices and strengthening legal enforceability. To allow for transferrable learning to other contexts, the following reflections will highlight the strengths of the Welsh approach along with a critique of it, to show how it can be improved further.

The first objective of this article was to review how radical legislative frameworks emerged, promoting children's rights and sustainable development through mechanisms aimed at embedding each agenda in administrative and political decision-making. This strong foundation is not without its concerns, and there is room for improvement, especially as to legal enforceability. The Senedd Children and Young People and Education Committee, the Children's Commissioner for Wales and children's sector organisations have argued that the impact of the due regard duty under the Rights of Children and Young Person's (Wales) Measure 2011 would be greater if extended to public bodies across Wales, and if a legal remedy were to be provided for a rights violation. Kilkelly recognises that 'legal implementation can be a gradual process and take different forms and paths depending on the national context', however, she insists that 'legal incorporation matters' (Kilkelly 2019, p. 322). Legislative reform has brought the CRC into the law of Wales, but the next step is to ensure that public bodies can be fully held to account.

The second objective was to examine the importance attached to children's participation in decision-making and some of the practices developed in Wales to support this. Through the Welsh Youth Parliament, child-led projects such as LBLV and the participative approaches adopted by public bodies, children have been successfully engaged in influencing decisions, including within the sustainable development agenda. These are steps towards addressing the concerns that 'children remain largely invisible through tokenistic and poorly executed approaches to their participation' (Davies 2019, p. 42). However, the challenge remains to sufficiently resource, support and facilitate these initiatives so they can be, 'meaningful for all concerned' (Arts 2019, p. 233). This requires further work to embed children's participation in the routine operations of public bodies, enabling localised and bottom-up approaches to contribute to learning on sustainability (Vandenhole 2019).

In this regard, the inclusion of Education for Sustainable Development and Global Citizenship education in the 2020 national curriculum is a significant step (Welsh Government 2020). Through such education, children's understanding and knowledge will increase and even exceed those of older generations, giving them much to contribute in the years to come. Therefore, it is even more important for public bodies to meet the challenges of operationalising, within their decision-making processes, children's right to be heard and to negotiate, within the entanglement of 'top-down' mechanisms, recognition of children's agency.

The third objective considered the extent to which the children's rights agenda and the sustainable development agenda have become better integrated in Wales, through guidance provided to public bodies. Kilkelly (2020) and Nolan (2019) both referred to the dissonance between the agendas at the international level and Vandenhole (2019), pointed out that a children's rights approach to sustainable development is still very much under construction. The very early steps taken in Wales towards converging the two agendas are innovative and potentially offer a route to integrating the discourse on SDGs and children's rights. However, much work is now needed to simplify the current complex matrix of objectives, targets and tools, to enable public body officials to fully understand and effectively implement them.

To conclude, the Wales case study suggests that accountability for children's rights and sustainable development should be further strengthened in law. It supports the argument that the two agendas are interconnected and need to be better aligned. Charting a path towards such alignment requires the development of a children's rights approach to sustainable development, including recognition of the crucial contribution of children themselves. Children should be recognised as autonomous rights holders and empowered to be active agents inspiring transformative change. This requires in practice, that guidance must be offered to those duty bearers who are responsible for implementation, accompanied by the necessary resources. The Wales case study offers approaches, tools and methods that may help to operationalise that goal.

**Author Contributions:** Conceptualization, R.C.; data curation, H.D., A.R., J.W.; formal analysis, H.D., A.R. and J.W. for the LBLV element of the research; funding acquisition, H.D., A.R., J.W. for the LBLV element of the research; investigation, R.C., H.D., A.R. and J.W. methodology, R.C., H.D., A.D., A.R., M.U. and J.W.; project administration, R.C., A.D. and M.U.; resources, R.C., H.D., A.D., A.R., M.U. and J.W.; supervision, R.C., A.D. and J.W.; validation, R.C., H.D., A.D., A.R., M.U. and J.W.; writing—original draft, R.C.; writing—review and editing, R.C., H.D., A.D., A.R., M.U. and J.W. All authors have read and agreed to the published version of the manuscript.

**Funding:** This research received no external funding.

**Institutional Review Board Statement:** Ethical review for all of the qualitative and quantitative data gathered from children involved in the LBLV project was secured from Swansea University School of Law Ethics Committee. The research was conducted in accordance with the Declaration of Helsinki.

**Informed Consent Statement:** Informed consent was secured from all child participants in all activities conducted by LBLV project.

**Data Availability Statement:** Data from LBLV can be found on the project's Lleisiau bach.org (accessed 3 March 2020) currently under re-construction.

**Conflicts of Interest:** The authors declare no conflict of interest.

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
