# Peer review of "Integrating Sustainable Development and Children’s Rights: A Case Study on Wales"

_socsci, doi:10.3390/socsci10030100_

Round 1

Reviewer 1 Report

This draft article has potential. It addresses an interesting topic and seeks to share experiences in Wales in the realm of governance for sustainable development and children’s rights. However, in my view, this piece would only become fully publishable after a rigorous revision, in which the following comments would be processed:

- The author(s) submitted their draft article to a general journal and thus need to consider that many readers will not be familiar with the Welsh context. Hence, throughout the article, they need to explain explicitly all issues relating to Welsh practices in such a way that they will be both understandable and appealing for a non-Welsh audience.

- It would be useful if, right at the start, the author(s) could explain a bit more clearly, and justify, the scope and objectives of the article. The abstract mentions 3 objectives: considering how recent guidance to Welsh public bodies for implementation attempts to contribute to a more integrated approach; discuss how the children’s rights agenda has empowered children to speak out on sustainable development; and to consider the ‘children as researchers’ methodology. It is not so clear what the combination of these elements will generate, in terms of answering the main question this article seeks to address. In addition, the reasons for picking up especially child participation and LBLV are not clear. What will the material on the latter (LBLV) add to the analysis, and why focus on a research methodology? Further down in the article, the author(s) refer to the objectives of informing policy (p. 3 line 81) and generating theory (p. 4 line 91). What are the main objectives of this article and what is the strategy for achieving them? Right at the start, the author(s) should introduce and justify his/her/their objectives and then pursue them systematically.

- If the article is a case study, the author(s) should state the following more explicitly, right at the start: what exactly does the case entail (period 1999-2020)?; what exactly was researched and for what purpose?; exactly what is this a case study of?; what could the case “Wales” mean/imply for other contexts? A rudimentary justification is provided at p. 3 lines 75-78. This could be stated earlier on in the text, and a bit more elaborately (with concrete issues/examples to make the claim a little more credible). If the case study element is so important, perhaps it should also be mentioned in the title of the article and in the abstract.

- The methodology section should explain more clearly why/how a case study on Wales could inform other people/issues in other country contexts. It should also explain how, concretely, the author(s) acted on her/his/their claim that they “acknowledge children as legitimate and autonomous rights holders” (p. 4 line 102). What did this mean for the research underlying this article and/or for writing up this article? At present this is not clear. In the next paragraph the author(s) state(s) that Welsh policy and legislation was analyzed, and international development discourses. How does this relate to children as legitimate and autonomous rights holders?

And, why was the ODI model (p. 4 line 112) selected? How does that model serve the author(s)’ purpose, and what where its implications for the research underlying this article and/or for writing up this article?

LBLV (p. 5 lines 116-117) needs more introduction as well.

In line with an earlier remark on this, the claim at p. 5 line 123 that the case study is “not generalisable” but “can offer transferable learning” will need to be at least somewhat substantiated.

- The whole text needs a careful final editing round. Various sentences do not run smoothly (e.g. p. 7, line 194-195; p. 8 lines 209-210; p. 24 line 639). Punctuation requires attention too. Writing in bullet points should be avoided as much as possible (e.g. p. 23). It leads to a text that reads too much like a summary instead of a properly explained line of argument.

- Referencing needs attention too. E.g., the first paragraph of section 3.1 (p. 5) needs to be referenced because it refers to various facts and figures. At page 10, the bullet points require page-specific references. The sources of the definitions provided in footnotes 53-55 should also be revealed. The citation at p. 10 lines 269-270 requires a page-specific reference. At p. 22 lines 578-592 need references too. The link provided in footnote 145 does not allow access to the Self Assessment Tool Kit or to CRIA (the links in the website do not work).

- Section 4.1. has a general human rights/HRBA focus. Why not provide more child rights/CRBA information?

- Sections 3 and 4 describe a lot of facts relating to Welsh legislation and policy. The article would become more interesting if the author(s) would weave more comments and/or critical and/or comparative remarks into the text. These would help the non-Welsh reader to appreciate this (fairly descriptive) material more.

- Long citations, such as at p. 17-18, should be avoided, unless they have explicit added value. In most instances, paraphrasing this kind of material will lead to a more concise, focused and appealing text which will more directly underpin the line of argument the author(s) wish to present. In this way, the child rights relevance of this general material might also come out better.

- The author(s) seem to suggest a few times (e.g. p. 3 line 63; p. 19 lines 501-503) that binding law, or enforceable human rights, would generate a better implementation record. However, not much evidence is mobilized to support this suggestion. Presumably, in Wales, the child rights included in the CRC are enforceable already. So the necessity and added value of more enforceable rights in the sphere of sustainable development and children is perhaps not self-evident. The authors should either fully substantiate this point or leave it out.

- At the end of section 4.4.1. (p. 22 line 588) it is not clear what the “great hope” is based on. Earlier on the author(s) described how a promising practice of child participation was discontinued once it had to be paid for. This is not a source of hope then, I would say.

- In section 4.4.2, it is not clear to me why LBLV gets so much attention. Is it the only prominent practice example? And, how is it relevant to sustainable development?

- Section 5 does not provide much insight into the practice since the adoption of “The Right way”, the self-assessment tool and CRIA. Has practice changed since December 2017? It would be nice to read practical examples of this, throughout the section.

- The conclusion can be improved and should be tuned to what a reader who is not familiar with the Welsh context should draw from the earlier sections. At page 31 lines 863-864 again emphasize the importance of a legislative framework (in line with the earlier statements about enforceable rights). But has this really helped in Wales? And, will it help in other contexts? This requires a more explicit argument and substantiation. In addition, the reference to COVID 19 at p. 32 line 871 underlines that law is ‘only’ one factor of what needs to be in place to realize sustainable development and children's rights. The very last line of the conclusion refers to “gains made” but in my opinion the author(s) have not yet explained sufficiently what the gains are (besides legislation). I guess the incidental (but discontinued) good practice of child participation would also qualify as gains. But what exactly is meant by “must not be lost”, and does this refer to Wales or to other places too?

Author Response

Dear Reviewer 1,

Many thanks for taking the time to review our manuscript and for your helpful advice on revisions. We include a grid below which outlines our response to your recommendations.

Reviewer Comment

Author Response

The author(s) submitted their draft article to a general journal and thus need to consider that many readers will not be familiar with the Welsh context. Hence, throughout the article, they need to explain explicitly all issues relating to Welsh practices in such a way that they will be both understandable and appealing for a non-Welsh audience.

The authors have aimed to explain throughout the article issues relating to Welsh practices.  We have also added further information and references in Section 2.1 to help readers to better understand devolution. We believe it is now accessible to readers who do not have a prior knowledge of Wales.

 It would be useful if, right at the start, the author(s) could explain a bit more clearly, and justify, the scope and objectives of the article. The abstract mentions 3 objectives: considering how recent guidance to Welsh public bodies for implementation attempts to contribute to a more integrated approach; discuss how the children’s rights agenda has empowered children to speak out on sustainable development; and to consider the ‘children as researchers’ methodology. It is not so clear what the combination of these elements will generate, in terms of answering the main question this article seeks to address. In addition, the reasons for picking up especially child participation and LBLV are not clear. What will the material on the latter (LBLV) add to the analysis, and why focus on a research methodology? Further down in the article, the author(s) refer to the objectives of informing policy (p. 3 line 81) and generating theory (p. 4 line 91). What are the main objectives of this article and what is the strategy for achieving them? Right at the start, the author(s) should introduce and justify his/her/their objectives and then pursue them systematically.

The authors have explained more clearly and justified, the scope and objectives of the article right at the start. Three objectives have been clarified and introduced systematically in the introduction. The emphasis on LBLV has been amended to make it simply one example of good participative practices that have emerged in Wales post-devolution and may contribute to the goal of converging practice in sustainable development and children’s rights. Please see Section 1. Introduction (pages 1-3)

If the article is a case study, the author(s) should state the following more explicitly, right at the start: what exactly does the case entail (period 1999-2020)?; what exactly was researched and for what purpose?; exactly what is this a case study of?; what could the case “Wales” mean/imply for other contexts? A rudimentary justification is provided at p. 3 lines 75-78. This could be stated earlier on in the text, and a bit more elaborately (with concrete issues/examples to make the claim a little more credible). If the case study element is so important, perhaps it should also be mentioned in the title of the article and in the abstract.

The explanation of the case study is made just after the international background in the introduction (Section 1) and lays out exactly what the case study entails via 3 objectives, including,

What was researched and for what purpose?

What exactly is this a case study of?

What could the case study Wales mean/imply for other contexts?

The case study as advised is mentioned in the title and the abstract (Pages 1-3).

The methodology section should explain more clearly why/how a case study on Wales could inform other people/issues in other country contexts. It should also explain how, concretely, the author(s) acted on her/his/their claim that they “acknowledge children as legitimate and autonomous rights holders” (p. 4 line 102). What did this mean for the research underlying this article and/or for writing up this article? At present this is not clear. In the next paragraph the author(s) state(s) that Welsh policy and legislation was analyzed, and international development discourses. How does this relate to children as legitimate and autonomous rights holders?

We have shortened the description on methodology and included it in the introduction (section 1, pages 1-3) simplifying it and putting it up front so the reader understands how we approached and conducted this research. It has been explained how the Wales case study can inform other people/issues in other country contexts.

The claim is also explained “acknowledge children as legitimate and autonomous rights holders” and how it informs the discussion on children’s participation in decision making (Section 1). The importance of children being recognised as rights holders and agents of transformative change in relation to both agendas is consequently referred to throughout the sections on children’s participation in decision making.

And, why was the ODI model (p. 4 line 112) selected? How does that model serve the author(s)’ purpose, and what where its implications for the research underlying this article and/or for writing up this article?

We believe this is too much detail for the reader. We have taken out because it disrupts the flow of this article and would require too much explanation or another article to do justice to it.

LBLV (p. 5 lines 116-117) needs more introduction as well.

We have reduced the prominence given to LBLV and refer to it as an example of a participative practice that has successfully contributed to decision making on sustainable development and children’s rights (see revised Abstract and Introduction)

LBLV has been given a more detailed introduction that aims to better contextualise its importance to the article and sustainable development. See Sec.3.4.1

In line with an earlier remark on this, the claim at p. 5 line 123 that the case study is “not generalisable” but “can offer transferable learning” will need to be at least somewhat substantiated.

We believe that this article is not about substantiating the value of qualitative case studies and discussing this. We have removed the word “generalisable” and just referred to the fact that the case study is transferrable to other contexts.

The whole text needs a careful final editing round. Various sentences do not run smoothly (e.g. p. 7, line 194-195; p. 8 lines 209-210; p. 24 line 639). Punctuation requires attention too. Writing in bullet points should be avoided as much as possible (e.g. p. 23). It leads to a text that reads too much like a summary instead of a properly explained line of argument.

Sentences have been edited.

Punctuation has been attended to.

Bullet points have been removed.

Referencing needs attention too. E.g., the first paragraph of section 3.1 (p. 5) needs to be referenced because it refers to various facts and figures. At page 10, the bullet points require page-specific references. The sources of the definitions provided in footnotes 53-55 should also be revealed. The citation at p. 10 lines 269-270 requires a page-specific reference. At p. 22 lines 578-592 need references too. The link provided in footnote 145 does not allow access to the Self Assessment Tool Kit or to CRIA (the links in the website do not work).

We have added in further references to Section 3.1 now 2.1.

We removed all the bullets as advised and placed definitions and the sources of the references to the footnotes. Now endnotes 49-54

The citation at previously page 10, we inserted a page specific reference. Now endnote 56.

We inserted more references to the LBLV section (previously page 22) all of the examples and the data will soon be able to be accessed at www.lleisiaubach.org . At the time of writing, data from the projects is in the process of being updated and re-visualised for publication.

The link in the footnote, now endnote 158 now allows access to the Self Assessment Tool Kit or to CRIA.

Section 4.1 has a general human rights/HRBA focus. Why not provide more child rights/CRBA information?

We have introduced what has become Section 3.1 with a more children’s rights based focus. 

Sections 3 and 4 describe a lot of facts relating to Welsh legislation and policy. The article would become more interesting if the author(s) would weave more comments and/or critical and/or comparative remarks into the text. These would help the non-Welsh reader to appreciate this (fairly descriptive) material more.

We have woven more critical remarks into what is now Section 2 and 3 to aid the non-Welsh reader.

Long citations, such as at p. 17-18, should be avoided, unless they have explicit added value. In most instances, paraphrasing this kind of material will lead to a more concise, focused and appealing text which will more directly underpin the line of argument the author(s) wish to present. In this way, the child rights relevance of this general material might also come out better.

We agree and the majority of long citations have been reduced.

The author(s) seem to suggest a few times (e.g. p. 3 line 63; p. 19 lines 501-503) that binding law, or enforceable human rights, would generate a better implementation record. However, not much evidence is mobilized to support this suggestion. Presumably, in Wales, the child rights included in the CRC are enforceable already. So the necessity and added value of more enforceable rights in the sphere of sustainable development and children is perhaps not self-evident. The authors should either fully substantiate this point or leave it out.

We agree that we may have not explained legal enforceability sufficiently, so we have done this in the text, to aid the reader to understand the point we are trying to make.

For example, it is a positive that we have the Rights Measure, but this offers a “due regard model”, this provides administrative and political accountability but it still does not offer a child the right to an individual remedy in a court of law or place a duty of due regard on public bodies.  We argue alongside other children’s rights advocates that for better enforceability that the law needs to be strengthened to a compliance model. Please see additional text in Section. 3.2. We also make further reference to this in the Conclusion Section 5.

We also believe that although a human rights enhancing piece of legislation WFGA 2015, there is no right to make an individual claim under the Act and there is no reference to human rights.  This weakens the legislation. This we believe is covered sufficiently by the discussion in Sec. 3.3

.

At the end of section 4.4.1. (p. 22 line 588) it is not clear what the “great hope” is based on. Earlier on the author(s) described how a promising practice of child participation was discontinued once it had to be paid for. This is not a source of hope then, I would say.

This is an important point, and we have substantiated this in the text (end of 3.4.1). The new youth parliament is firmly situated on the parliamentary side of governance, it can better hold government to account and it is also embedded into the Welsh Parliament’s Commission which means it is not vulnerable any more like the previous charitable structure of Funky Dragon which was funded and overseen by the executive.

In section 4.4.2, it is not clear to me why LBLV gets so much attention. Is it the only prominent practice example? And, how is it relevant to sustainable development?

As part of the introduction to LBLV (Sec.3.4.2.) we have more clearly explained why we chose to focus on LBLV, its contribution to the sustainable development agenda and its ability to support children as autonomous rights holders.

- Section 5 does not provide much insight into the practice since the adoption of “The Right way”, the self-assessment tool and CRIA. Has practice changed since December 2017? It would be nice to read practical examples of this, throughout the section.

This was not part of the research, as the Right Way is still in early stages of implementation. However we have inserted some of the practical examples of embedding children’s rights approach to sustainable development as outlined by The Right Way, to explain to the reader some of the public body approaches at the end of Section 4.

The conclusion can be improved and should be tuned to what a reader who is not familiar with the Welsh context should draw from the earlier sections. At page 31 lines 863-864 again emphasize the importance of a legislative framework (in line with the earlier statements about enforceable rights). But has this really helped in Wales? And, will it help in other contexts? This requires a more explicit argument and substantiation. In addition, the reference to COVID 19 at p. 32 line 871 underlines that law is ‘only’ one factor of what needs to be in place to realize sustainable development and children's rights. The very last line of the conclusion refers to “gains made” but in my opinion the author(s) have not yet explained sufficiently what the gains are (besides legislation). I guess the incidental (but discontinued) good practice of child participation would also qualify as gains. But what exactly is meant by “must not be lost”, and does this refer to Wales or to other places too?

As advised we made significant changes to the conclusion so that it links more clearly with the three objectives as outlined in the introduction and the points made in the discussion, to more helpfully assist the reader with transferrable learning to their own context. See Section 5.

Reviewer 2 Report

I consider that the manuscript is relevant but it is necessary to undertake certain recommendations for its improvement, such as:

-In the introduction, make sure the paragraphs are connected. Sometimes the ideas are out of order and do not capture the reader's attention.

-It is necessary to clarify the method and procedure used. Add more information. These are very broad sections that do not clarify the processes followed. The procedure for the selection of participants (inclusion and exclusion criteria) has not been clarified.

-Indicate in which language the instruments were.

-Clear regression tables.

-It is recommended to clarify the discussion section. The section needs to be more coherent and to provide clarity to the reader.

-It is necessary to improve the conclusions section. Clarify presented ideas more consistently.

-It is recommended to use recent citations to support the study.

Author Response

Dear Reviewer 2,

Many thanks for taking the time to review our manuscript and for your helpful advice on revisions. We include a grid below which outlines our response to your recommendations.

Reviewer 2 Comments

Authors Response

In the introduction, make sure the paragraphs are connected. Sometimes the ideas are out of order and do not capture the reader's attention.

The authors have explained more clearly and justified, the scope and objectives of the article right at the start. The objectives have been clarified and introduced systematically in the introduction to capture the reader’s attention. Please see Section 1. Introduction.

It is necessary to clarify the method and procedure used. Add more information. These are very broad sections that do not clarify the processes followed. The procedure for the selection of participants (inclusion and exclusion criteria) has not been clarified.

We have clarified the method and procedure in the introduction as aligned to our 3 objectives.

We clarify the selection of participants in section 3.4.2 paragraph 5.

Indicate in which language the instruments were.

The research with child participants was conducted in English and Welsh language. See 3.4.2 paragraph 5.

Clear regression tables.

We did not use regression tables.

It is recommended to clarify the discussion section. The section needs to be more coherent and to provide clarity to the reader.

We have made this section clearer to provide clarity to the reader (sections 2-4).

It is necessary to improve the conclusions section. Clarify presented ideas more consistently.

As advised we rewrote and improved the conclusion to align more clearly with the 3 objectives in the introduction and to assist transferrable learning to other contexts. Please see Section 5.

It is recommended to use recent citations to support the study.

We used a range of recent references. Little has been written in the area of children’s rights and sustainable which is what makes this Special Edition such an important contribution to the knowledge base. We strongly referenced throughout a recent 2019 book edited by Feyton Glyn, on Children’s Rights and Sustainable Development, the only English language book that has been written on this subject to date.

With reference to citations we used citations that were time and context dependent, to help explain the evolving case study over the 20 years of devolution. However we reduced a number of citations throughout the manuscript to help the flow of the text.

Round 2

Reviewer 1 Report

The author(s) have processed feedback quite well. As a result of the revisions made, the article has gained in clarity and rigour. In addition, it became more accessible for readers from other contexts than Wales and allows them more easily to draw insights from the Welsh experience that might be relevant for their own context. I certainly find the piece publishable now.

Reviewer 2 Report

I believe that the authors have made important changes to the manuscript and that this version has improved.